# Culturing Important Plants for Sweet Secondary Products under Consideration of Environmentally Friendly Aspects

**Matthias Preusche [1,2], Andreas Ulbrich [1,*] and Margot Schulz [2,*]**

1   Department of Horticultural Production, University of Applied Science, 49090 Osnabrück, Germany; m.preusche@hs-osnabrueck.de
2   Institute of Molecular Physiology and Biotechnology of Plants (IMBIO), University of Bonn, 53127 Bonn, Germany
*   Correspondence: a.ulbrich@hs-osnabrueck.de (A.U.); ulp509@uni-bonn.de (M.S.); Tel.: +49-(0)-541-969-5116 (A.U.); +49-(0)-732151 (M.S.)

**Abstract:** Some sweet tasting plant secondary metabolites are non-caloric or low nutritive compounds that have traditional use in food formulations. This mini-review focuses on conventional and advanced cultivation regimes of plants that accumulate sweet tasting or sweet taste modulating secondary metabolites of potential economic importance, in particular mogrosides (*Siraitia grosvenorii*), phyllodulcin (*Hydrangea macrophylla*), glycyrrhizin (*Glycyrrhiza glabra*), steviol glycosides (*Stevia grosvenorii*), and rubusoside (*Rubus suavissimus*). Consequential obstacles during the cultivation of *Hydrangea macrophylla* cultivars outside their natural habitat in a protected cultivation environment are addressed. Culturing at non-habitat locations facilitates short transportation routes of plant material for processing, which can be a key to an economically and environmentally compatible usage. The biosynthetic pathways, as far as known, are shortly mentioned. The proved or hypothetical degradation pathways of the compounds to minimalize environmental contamination are another focal point.

**Keywords:** *Siraitia grosvenorii*; *Hydrangea macrophylla*; *Glycyrrhiza glabra*; *Stevia rebaudiana*; *Rubus suavissimus*; culture systems; mogrol; phyllodulcin; glycyrrhizin; steviol

## 1. Introduction

Sweet tasting secondary compounds belong to very different chemical classes, for instance, lactones and phenolic compounds, flavonoids, terpenoids, and saponins as well as proteins. Only a few natural sweet tasting molecules (glycyrrhizin, steviol) currently have wider use, accounted for by laborious culture conditions as the plants demand special requirements for their growth, or the lack of simple procedures for compound isolations. In addition, the robust elucidation of molecular and developmental backgrounds that trigger the compounds' biosynthesis and accumulation modes of the producing plants are mostly missing. Therefore, many principally interesting natural sweet tasting secondary plant metabolites are presently of restricted value to the market. As the global market for sweet tasting metabolites is expected to grow significantly during the next few years, it is necessary to increase the knowledge regarding the regulation of the biosynthesis and production systems through additional research and to develop a method that guarantees high yields of the relevant metabolites. This mini-review mainly centers on plant species with increasing importance as sources for natural sweet tasting molecules, (*Glycyrrhiza glabra*, *Stevia rebaudiana*). In addition, three other species that have a broader use predominantly in China are addressed, *Siraitia grosvenorii*, *Rubus suavissimus*, and *Hydrangea macrophylla*, which originated from Japan. While the culture systems are one topic, the biosynthesis of the compounds is—as far as known—also addressed. The second topic presents known detoxification products and proved or hypothetical degradation via microbial activities, as sustainability and degradability of the compounds are of outstanding importance. A non-residue degradation, one of the Green Chemistry Principles (ACS), is a prerequisite

to avoiding environmental contaminations. In this context, also short transportation routes for the trading goods have to be considered.

## 2. Plants

### 2.1. Siraitia grosvenorii (Swingle) C. Jeffery, Luohanguo

*S. grosvenorii* (Cucurbitaceae) is native to parts of China and Thailand. The fruits contain cucurbitane triterpenes named mogrosides, possessing a sweetening potency that is up to 500 times higher than that of sucrose [1,2]. For *S. grosvenorii*, a rise in market value is expected despite the existing difficulties with the culture.

### 2.2. Hydrangea macrophylla Seringe var. Thunbergii Makino (Saxifragaceae)

*Hydrangea macrophylla* can be cultured in the temperate climates of Europe, Asia, and North America. *Hydrangea macrophylla* contains about twenty different dihydrocoumarins, among them the sweet tasting phyllodulcin [2]. Phyllodulcin is mainly extracted from the leaves. In certain *H. macrophylla* cultivars, the contents of the biosynthetic precursor hydrangenol have increased more than tenfold during the last few years [3]. The reason for the increase needs yet further elucidation. Hydrangenol may have undesirable application properties and is difficult to separate from phyllodulcin during the extraction. For commercial use of the leaves, a minimal content of hydrangenol is, therefore, aspired.

### 2.3. Glycyrrhiza glabra L. (Fabaceae), Licorice

*Glycyrrhiza glabra* L. is an herbaceous perennial native to areas with a temperate to subtropical climate in Europe and Asia. Important culture regions are in China, India, and Pakistan. The root contains glycyrrhizin, an oleanane-type triterpenoid saponin, decorated with $O$-β-d-glucuronosyl-(1′→2)-β-d-glucuronic acid. The compound is 50–200 times sweeter than sucrose but has a specific, undesired aftertaste. The monoglucuronide is sweeter than the parental compound [2,4]. Glycyrrhizin is also synthesized in other *Glycyrrhiza* species, for instance, *G. uralensis*. In Asia, particularly in China and Japan [5], the use of licorice has a long tradition as a medicinal plant and as a food additive also in Europe. The compound can cause health problems (pseudoaldosteronism) when consumed in large quantities [6]. On the other side, glycyrrhizin is a pharmacologically interesting compound with efficiency against HIV- and SARS-related viruses and possesses many other beneficial medicinal properties [7,8].

### 2.4. Stevia rebaudiana Bertoni

The Asteraceae species *Stevia rebaudiana* Bertoni, an herbaceous perennial, is native to South America. Presently, the plant is cultured in areas with a suitable climate in Central America and Asia [1]. The plant is not frost resistant. In the leaves, numerous kauran-type diterpenoids glucosides accumulate (rebauside A-F, stevioside, steviolbioside, and dulcoside A). Several of the glucosides have a sweet taste with a bitter aftertaste [2], with stevioside and rebaudioside A as the most abundant ones. The total concentration of the compounds varies from 4 to 20%/per leaf dw. Steviol glycosides are 200 to 400 times sweeter than sucrose [1]. Steviol glycosides have long been used in Brazil and Japan and were approved in Europe, the United States, and Canada about ten years ago [5].

### 2.5. Rubus chingii v. suavissimus S. Lee (Rosaceae)

*Rubus chingii v. suavissimus* or *Rubus suavissimus*, native to Southern China, contains rubusoside in its leaves, a steviol-bisglucoside that is about 100-fold sweeter than sucrose. *R. chingii v. suavissimus* is the only *Rubus* species, which accumulates diterpenes [2].

## 3. Culture Systems

### 3.1. Siraitia grosvenorii

As stated by Shivani et al. [9], good conventional agricultural practices are not available, the species has poor adaptability and problematic pollination, and germination and

plants raised from seed show inherent heterozygosity. The current research gives hope for progress toward a more successful culture from seeds, but approaches to establishing tissue culture methods were assumed to be more promising for plantlet mass production. However, the establishment of tissue culture methods disclosed disadvantages as well [10–12]. Traditional propagation from cuttings increased viral infections, accompanied by low survival of plants obtained by micropropagation in the field. In addition, photoautotrophic micropropagation and ex vitro rooting methods did not constitute a satisfactory breakthrough. Temporary immersion system culture using single-node micro-cuttings, as described by Yan et al. (2010), seems presently to be the most promising method for obtaining plantlets suitable for field production. Today, *S. grosvenorii* is cultured mainly in China and perhaps increasingly in India [9,13]. Mogrosides, however, actually cannot legally be used as sweeteners in Europe or the United States [1], despite being used in China for hundreds of years.

### 3.2. Hydrangea macrophylla

*H. macrophylla*, used as an ornamental plant, is cultured either from seeds or cuttings or by in vitro propagation; the latter includes micro- and macro-propagation methods. In 2017, Arafa et al. [14] published a method for large-scale in vitro propagation based on sterilized shoot tips and nodes. Explants were cultured in MS nutrient medium containing sucrose (30 g/L). The multiplication of well-developed shoots was performed as described by Sacco et al. [15]; rooting was induced, for instance, with IAA. The finally obtained plantlets were transferred into pots with an optimized soil mix (peat moss/sand, 3:1 *v/v*), for adaption in the greenhouse. Further, Ruffoni´s group established a protocol for temporary immersion shoot culture with *Hydrangea* genotypes [16].

However, for the profitable extraction of phyllodulcin, plants accumulating high amounts of phenyldihydroisocoumarins are needed. The establishment of conventional culture systems for phyllodulcin production was recently attempted at the University of Applied Sciences in Osnabrück, with the aim to develop cultivation regimes that enable the plants to synthesize high amounts of phyllodulcin throughout the year [3] (Figure 1). The propagation of such plants is obtained by cuttings from suitable mother plants. Rooting of the cuttings and the subsequent cultivation of the young plants is possible according to the common methods described for ornamental *H. macrophylla* species [17,18]. Cuttings are usually taken early in the year—in Germany, preferably between February and May, at the latest in June. Rooting starts under normal greenhouse conditions in small pots or trays, using a peat substrate with low salt content.

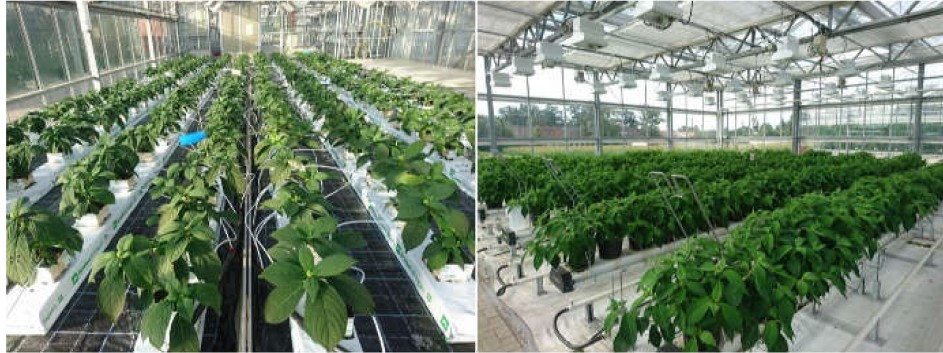

**Figure 1.** *H. macrophylla* grown in greenhouses using two different substrates for hydroponic cultivation: rock wool (**left**) and perlite (**right**).

It is also possible to cultivate the species in hydroponic systems (Figure 1). Here, frequent pruning maximizes the leaf biomass production under optimal growth conditions. The extractable phyllodulcin content can be increased by this method.

As mentioned, most *H. macrophylla* cultivars are ornamental plants, mainly bred for better flowering behavior [19], improved growth properties, or cold hardiness [20], but not

for high phyllodulcin contents. Therefore, it is necessary to screen the cultivars for their phyllodulcin contents and select elite cultivars. Suitable genotypes with relatively high amounts of phyllodulcin are native to Japan [21].

There are reliable ways to enhance the amounts of extractable phyllodulcin. The amount in inflorescences is considerably lower than in leaves. In leaves, the content is highest in younger leaves and declines in older leaves (Figure 2). An increase in the phyllodulcin content can be achieved by the fermentation of the leaves after harvesting. Here, enzymatic hydrolysis releases phyllodulcin from its glucosylated precursor, presenting a method that has been common in Japan for centuries. The application of methyljasmonate (MeJA) directly on the leaves enhances the phyllodulcin pool as well. Since hormone treatment may only transiently elevate phyllodulcin levels, applications should be done prior to harvesting. Interestingly, the application of MeJA significantly increases the phyllodulcin content in the leaves of vegetative plants (Figure 3), but the same treatment does neither elicit a comparable nor consistent effect in plants during the generative growth phase.

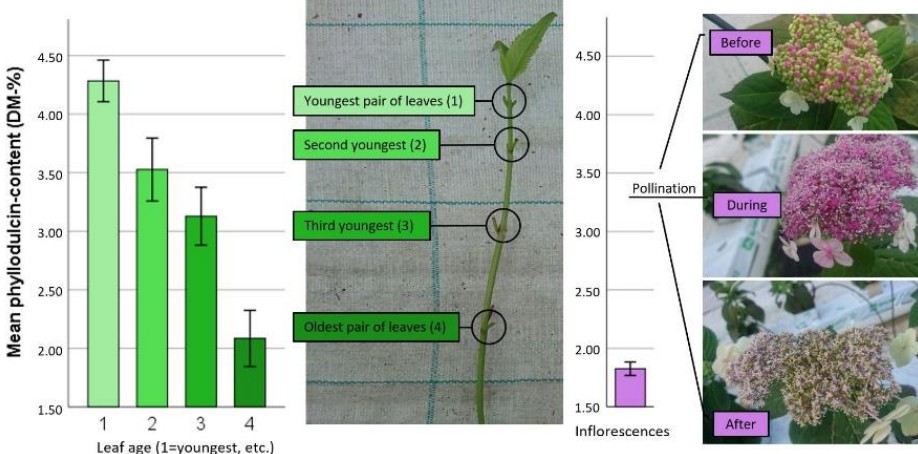

**Figure 2.** (**Left**) Average phyllodulcin content in leaves of different ages (*n* = 15) and (**right**) in inflorescences of different developmental stages during anthesis (*n* = 8, composite samples; error bars: 95% CI).

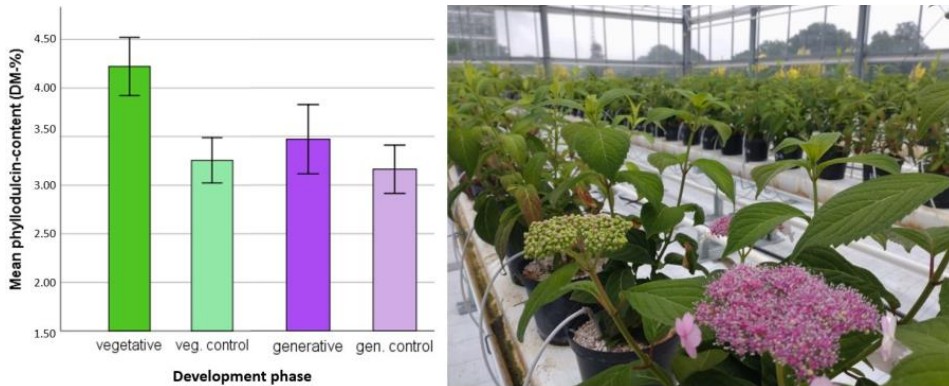

**Figure 3.** (**Left**) Mean PD content (dry-matter percentage) in leaves of vegetative and generative *H. macrophylla* (*n* = 72) after several MeJA applications (error bars: 95% CI). (**Right**) Plant culture exhibiting vegetative and generative growth in the same greenhouse.

Furthermore, flowering is unfavorable for phyllodulcin enhancement, as it is accompanied by a cessation of growth, leading to reduced biomass production (Figures 2 and 3). Plants cultured as shown in Figure 1 tend to switch to the generative phase in late summer, cease growing, and become dormant, thereby rendering a year-round profitable phyllodulcin production impossible [22]. The dormancy behavior and intricacies of flowering induction are well studied for ornamental *H. macrophylla*. The breaking of dormancy can

be achieved only by exposing the plant to cold temperatures, preferably below 5 °C for about 1000 h [17,18,23].

Many *H. macrophylla* cultivars induce their flowers under short day conditions [3]; thus, long daylight regimes may help to prevent flowering. While most species induce their flowers in autumn and, after dormancy, start flowering in the early summer when the days get longer, there are cultivars that behave differently to common stimuli influencing flower induction [24]. Some cultivars have the ability to flower despite losing their buds in the winter. This remontant flowering potential is observed with several of the commercially produced *H. macrophylla* cultivars [25]. Thus, *H. macrophylla* shows multifaceted responses to growth stimuli, resulting in an enormous variation of the growth properties, even among the well-established cultivars.

Periodic pruning is the most common method to prevent the generative growth of mother plants used for the production of cuttings [26]. Since the Japanese wild plants behave similarly to remontant cultivars, at least when grown outside their natural habitat, periodic pruning is inevitable in order to maintain vegetative growth. There is no reliable horticultural practice to promote vegetative growth once generative growth sets in. Rigorous pruning, repeated removal of flower buds, and high ambient temperatures (>25 °C) in addition to long daylight regimes (>16 h) are not sufficient to suppress flowering and promote vegetative growth. [27]. In consequence, the growth rate will continue to decrease until the plants are dormant. Furthermore, after dormancy has passed, the Japanese cultivars will immediately continue to resume generative growth, even if the flower buds have been removed [28]. Hence, it is crucially important that pruning is done frequently and before generative growth sets in.

### 3.3. Glycyrrhiza glabra

Culture systems have been developed mainly for pharmaceutical purposes. Since stolon division leads to low progeny and culture from seeds is problematic due to low germination rates, conventional propagation seems not to be the method of choice for the large-scale production of plantlets [29]. Therefore, several in vitro propagation methods have been developed to improve plant material for glycyrrhizin accumulation [7,29–31]. Shaheen et al. [29] described an efficient micropropagation method for *G. glabra*. Intermediate nodal explants were most suitable for establishing aseptic cultures. High bud break and shoot proliferation were obtained with MS medium containing up to 3 mg L$^{-1}$ thidiazuron and 6-benzyladenine, while MS media with naphthalene acetic acid at 6 mg L$^{-1}$ was optimal for rooting. By this method, healthy plantlets were generated that could be easily acclimated to ex vitro conditions. The development of in vitro culture systems has been highly important for many years because wild harvesting brought *Glycyrrhiza glabra* to the brink of extinction, while the medical use of glycyrrhizin is increasing worldwide.

Nodal explants have prevailed for axillary shoot proliferation, and shoot tip culture and apical and axillary buds or in vitro stolon culture systems have been used for direct regeneration, and presently, tissue culture technology is used for the mass production of an elite population [29,30]. An optimized culture regime for callus initiation and multiplication for plant regeneration was developed by Rathi et al. [31]. Srivastava et al. [7] developed a hairy root culture method for glycyrrhizin production. These cultures were established from leaves of precultured shoots by the use of an infection medium supplemented with *Agrobacterium rhizogenes* A4 strain.

The recent identification of the total biosynthetic pathway and the responsible genes now opens the possibility of developing biotechnological processes for glycyrrhizin production, allowing for bypassing in vitro culture systems and protecting the endangered wild plant species [32]. Chung et al. [32] produced the *Saccharomyces cerevisiae* strain GL0–3 for the novo synthesis of glycyrrhizin; thus, a future plant-independent production of the compound could be in sight.

### 3.4. Stevia rebaudiana

The conventional propagation of *Stevia* is done by seeds or vegetative cuttings. However, the germination rate of the seeds, as well as propagation from cuttings, is low and results in plants with high variations in the stevioside contents. Therefore, in vitro propagation is preferred, while node explants are the common plant material used for micropropagation. Tissue culture systems in liquid medium were found to be a practicable method, but the immersion systems lead to physiological malformations, resulting in plants with low mechanical stability and other negative properties. Temporary immersion systems mitigate or even prevent the problems. Such systems have been developed by many authors ([33] and mentioned articles therein) by the use of a RITA® bioreactor with defined immersion period applications. Temporary immersion systems performed in the bioreactor lead to healthier plants. Ramírez-Mosqueda et al. [33] also found a low genetic variation according to ISSR analysis. However, the amount of stevioside was not determined. Environmental conditions, such as the light regime, temperature, possible microbial interactions, and mycorrhization modify the steviol glucoside biosynthesis. According to Libik-Konieczny et al. [34], the regulation of steviol glucoside metabolism is not yet understood. In addition to temporary immersion systems, the metabolic engineering of microorganisms, such as *E. coli* or *Saccharomyces cerevisiae*, for compound production will gain importance when all the genes necessary for the biosynthesis of a defined compound are known [35]. For steviol glucosides, the biosynthetic steps, including the enzymes and encoding genes have been identified [35,36] Moon et al. [36] described the construction of a genome-engineered *E. coli* strain able to produce 38.4 ± 1.7 mg/L steviol in batch fermentation. According to a scientific opinion, there are no safety concerns for steviol glycoside preparations in genetically modified strains of *E. coli* K-12. The resulting, highly pure compounds are recommended for use as a food additive in Europe [37].

### 3.5. Rubus suavissimus

The plant is mainly cultured in China, where it has been used for medical purposes since ancient times. Tissue culture systems from stem explants are under investigation (Lin Rong, Wang Run-Zhea, and Wang Xiu-Qin. Guaugxi Institute of Botany: Studies on tissue culture of *Rubus suavissimus*; note, in Chinese). Unfortunately, rubusoside extraction from plant material is not efficient. More efficient methods are the heterologous gene expression of glycosyltransferases for steviol glycosylation or the production of rubusoside from Stevia extracts by enzymatic conversion using suitable glycosidases (Figure 4). Yan et al. [38] isolated a β-glucosidase from *Chryseobacterium scophthalmum* 1433, which converted 99% of the applied steviol glycosides into rubusoside. Industrial applications of these methods are future aims that will obviate the need for *Rubus suavissimus* plant culture systems.

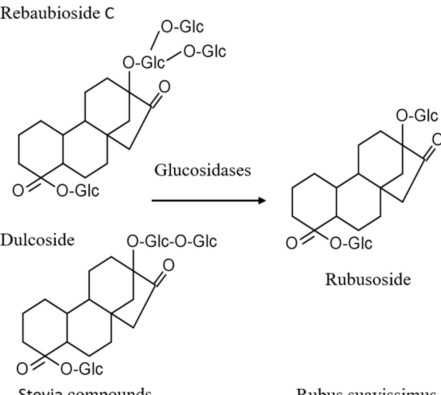

**Figure 4.** Rubuside production from *Stevia reaudiana* compounds.

## 4. Biosynthesis and Degradation of Mongrosides, Phyllodulcin, Glycyrrhizin, and Stevioside

### 4.1. Mogrosides

Mogrosides do not accumulate in vegetative parts of the plant but only in the fruit. The biosynthesis of the compounds was intensively investigated. The genes involved in biosynthesis were found to exhibit a highly coordinated expression pattern [39]. Mogrosides contain the cucurbitane triterpene skeleton, as do the bitter tasting and toxic cucurbitacines, characteristic secondary metabolites of the Cucurbitaceae. However, in the case of mogrol biosynthesis, the precursor 2,3 oxidosqualene is subsequently differently substituted with oxygen containing functional groups yielding tetra-hydroxycucurbitane (mogrol). Cucurbitadienol synthase catalyzes the formation of 24, 25 epoxy cucurbitadienol, which is transformed into mogrol by epoxidhydrolase. The following glucosylation steps yielding mogroside V are performed by glucosyltransferases. Thus, the last steps in biosynthesis encompass glucosylation reactions, leading to a group of mogrolglucosides (mogrosides) that differ in the number of attached glucose moieties. Deglucosylation results again in mogrol. (Figure 5).

**Figure 5.** Biosynthesis of mogroside V in *Siraitia grosvenorii* and hydrolysis of consumed mogroside in the gut of humans (simplified).

The metabolism of mogrosides was studied in humans and rats. In humans, mogrosides are only minimally absorbed, while the majority is excreted after deglucosylation in the gut. Glucose hydrolysis is either complete and results in mogrol or it is incomplete, yielding mono- and diglucosides [40]. In the feces of rats, the same metabolites were identified, while in portal blood mogrol, the monoglucoside and its glucuronic acid or sulfate conjugates were found [41]. Thus, mammalian intestinal microorganisms, also when acting in concert, are not able to degrade the triterpene skeleton. Triterpenoid catabolism is not well investigated. He et al. [42] assumed that combinations of photodegradation, microbial conversions, and reductive and oxidative processes will finally destroy the triterpenoid structures, generating polymers and aromatic ring systems, where the latter can be destroyed by a number of soil bacteria [43]. Since triterpenes of plant origin and some oxidized forms can be found in sediments, the complete degradation of triterpenes seems to be problematic in nature [44,45].

### 4.2. Phyllodulcin—Biosynthesis and Degradation

The biosynthesis of phyllodulcin is not fully understood, and only a few investigations have been performed to elucidate the pathway. Precursor feeding studies substantiate the assumption that 4-coumaroyl and dihydro-4-coumaroyl are precursors for hydrangeic and lunularic acid but do not lead to phyllodulcin accumulation. Stilbenecarboxylate synthase (STCS) is probably responsible for lunularic acid synthesis [46]. Çiçek et al. [2] hypothesized alternative biosynthetic routes for the phenyldihydroisocoumarin thunberginol G

biosynthesis, all starting from the phytoalexin resveratrol. This stilbene is formed by the condensation of p-coumaroyl-CoA with three malonyl CoA units [47]. However, also by considering the hypothesized alternative biosynthetic routes, the biosynthetic sequence leading to phyllodulcin remains elusive.

While biosynthetic intermediates of phyllodulcin are obscure, metabolites occurring during the compound´s degradation in animals have been identified. Interestingly, thunberginol G and hydrangenol, compounds that are suspected as being anabolic intermediates, also belong to the catabolic product assembly.

Several urinary and fecal metabolites have been identified when rats were fed with phyllodulcin [48], such as phyllodulcin-3′-*O*-sulfate, phyllodulcin-3′-*O*-β-glucuronide, 2-[2-(3,4-dihydroxyphenyl)ethyl]-6-hydroxybenzoic acid, 2-[2-(3-hydroxy-4-methoxyphenyl)ethyl]-6-hydroxy benzoic acid, thunberginol G, and hydrangenol. The benzoic acids and thunberginol G are produced in the liver and by gut-colonizing microorganisms, and hydrangenol only in the liver. Although not further investigated by Yasuda et al. [48], the aromatic systems of the catabolic benzoic acids might be completely degraded by coworking microorganisms that are able to perform aromatic ring cleavage and oxidation of the breakdown products in the TCA cycle (Figure 6).

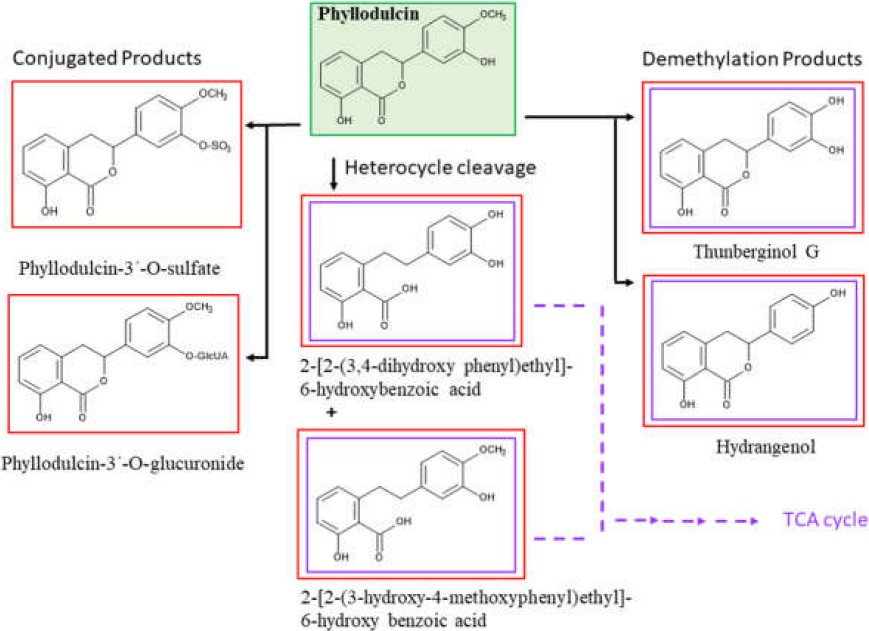

**Figure 6.** Detoxification and degradation of phyllodulcin in mammals (red outlines/black arrows) and breakdown of the aromatic ring system by bacteria (purple outlines/lines/arrows; dashed line: hypothetical). Green outlines/box: plant originated compound.

For complete degradation by bacteria, it is probably necessary to convert all phyllodulcin first into benzoic acids. We hypothesized a similar degradation pathway of 2-[2-(3,4-dihydroxy phenyl)ethyl]-6-hydroxybenzoic acid, 2-[2-(3-hydroxy-4-methoxyphenyl)ethyl]-6-hydroxy benzoic acid, as was recently described for resveratrol by *Acinetobacter oleivorans* strain JS678 [49]. In this case, the compounds would be oxidatively cleaved into simple phenolic acids by an enzyme comparable to *A. oleivorans* strain JS678 resveratrol oxygenase, followed by enzymatic conversions into molecules that are substrates for bacterial dioxygenases. The resulting cleavage of the aromatic system leads to breakdown products able to enter the TCA cycle [43]. Although the assumed degradation pathway still has to be proved in detail, phyllodulcin and its degradation metabolites may not accumulate in the environment in problematic concentrations.

### 4.3. Glycyrrhizin—Biosynthesis and Degradation

Precursors for biosynthesis are available from the MEP pathway in the chloroplast. The biosynthesis of glycyrrhizin starts with the cyclization of 2,3-oxidosqualene, yielding β-amyrin as the first compound of the sequence with the triterpene structure (Figure 7). The reaction is catalyzed by β-amyrin synthase and the following site-specific oxidations by cytochrome P450 monooxygenases, which leads to aglycon glycyrrhetinic acid. The aglycon is glucuronylated by an ER-located cellulose synthase superfamily-derived glycosyltransferase, which was recently characterized, and by a glucoronyltransferase to glycyrrhizin. The second glucuronylation is performed by glycosyltransferase GlcAUGT [32].

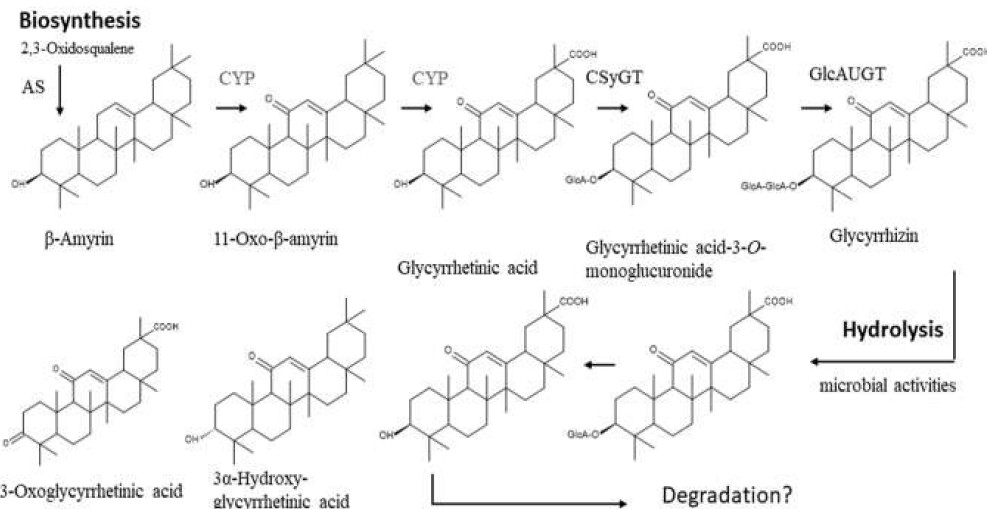

**Figure 7.** Biosynthesis of glycyrrhizin in *Glycyrrhiza* spec. and hydrolysis of glycyrrhizin to aglycon glycyrrhetinic acid in the human gut (simplified).

In the human gut, glycyrrhizin is converted again to aglycon by microbial activities. The major metabolite is aglucon, a minor glycyrrhetinic acid-3-*O*-β-D-glucuronide, together with traces of 3-oxoglycyrrhetic acid and 3α-hydroxyglycyrrhetic acid. The trace compounds are re-converted to the parental aglycon [50], which may be the reason why there seems to be no other report on their transient existence. Presently, only glycyrrhizic acid was described to be the bioactive compound that influences the gut microbiome. The aglycon is absorbed after ingestion and occurs in urine and in feces for excretion. The difficulties with subsequent microbial triterpene degradation in nature are mentioned above. Many fungi and some bacteria are able to modify triterpenes by a variety of enzymatic catalyses [51,52], but an efficient, natural pathway for the total breakdown of the triterpene scaffold is hitherto not known.

### 4.4. Steviol—Biosynthesis and Degradation

Geranylgeranyl diphosphate from the MEP pathway is converted to ent-kaurene in the chloroplasts. The step directed to the steviol synthesis is the conversion of geranylgeranyl diphosphate into *ent*-copalyl diphosphate, catalyzed by copalyl diphosphate synthase. The following cyclization to *ent*-kaurene is performed by kaurene synthase. *ent*-Kaurene is oxidized and hydroxylated to steviol by kaurene oxidase and kaurenoic acid 13-hydroxylase [35,36]. ER-located oxidases (yellow box, Figure 8) produce ent-kaurenoic acid, hydroxylated by kaurenoic acid 13-hydroxylase into steviol, which is released into the cytosol. In the cytosol, four different UDP glycosyl transferases are responsible for the final glyco-decoration of the aglycon, resulting in steviolmonoside, steviolbioside, stevioside, and rebauside A [35]. The biosynthesis occurs only in green tissue and involves different compartments, namely the chloroplasts, ER, and cytoplasm. The glycosides are stored in the vacuole [34]. Certain microorganisms degrade steviol completely via the intermediates monicanol and monicanone (Figure 8).

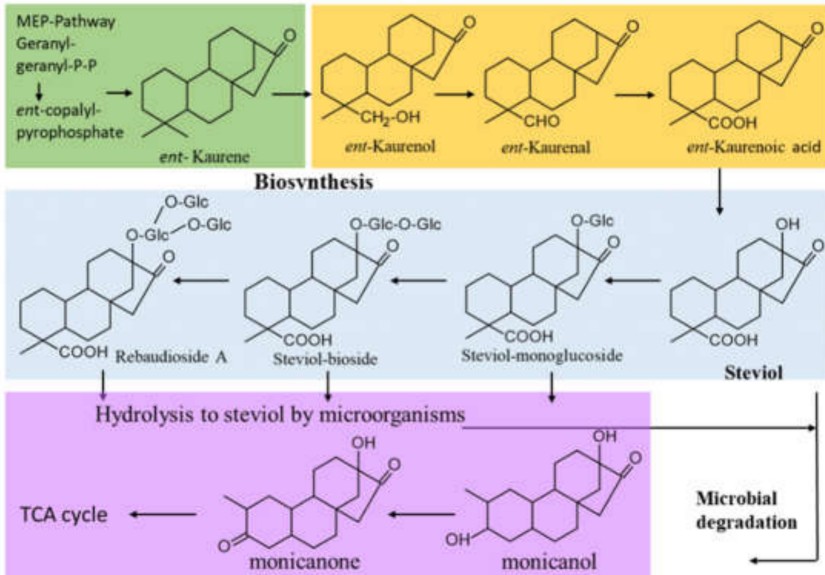

**Figure 8.** Biosynthesis and degradation of stevioside (simplified). Chloroplasts: green box. ER-located oxidases: yellow box. Cytosol: blue box. Degradation: purple box.

The metabolization of stevioside was investigated in mice, rats, hamsters, and humans by Hutapea et al. [53]. In these organisms, stevioside was hydrolyzed into steviol by intestinal microorganisms (Figure 8). The aglycone is absorbed, transported to the liver, then glucuronidated and excreted in the urine. When not absorbed, it is excreted in the feces [54,55]. In humans and mice, but not in rats and hamsters, steviol-16,17$\alpha$-epoxide was found as an intermediate, which is reconverted into steviol. However, steviol-16,17$\alpha$-epoxide is not mentioned by other research groups. Since a much higher reactivity of the epoxide has to be expected, one can speculate that some of the trace compounds may also form adducts or other products with biomolecules and can, therefore, not be captured by the common analytical methods. In organisms, no further degradation or detoxification product has been found.

The diterpenoid steviol glucosides were very recently found to be biodegradable [55]. Paraguayan soil bacteria are able to degrade the glycosides completely via the intermediates monicanol and monicanone under aerobic conditions (Figure 8). However, except for the two oxidation steps leading to monicanol and monicanone, subsequent catabolism resulting in the total molecule breakdown is still unknown. Nevertheless, the work of Meesschaert et al. [55] designates steviol compounds as sweet tasting compounds that fulfill the Green Chemistry Principles of non-residue degradation. There might be no risk of groundwater contamination when microorganisms able to perform the degradation are not overburdened.

## 5. Discussion and Conclusions

The increasing requirement for natural compounds with influence on sweet taste perception presents a challenge for sustainable production in saving the environment, wild plants, and their habitat. The development of tissue culture technologies and, when biosynthetic pathways and genes are known, genome-engineered microorganisms for production in bioreactors are promising approaches to fulfill the principles of green technologies, that is, there is no release of compounds from synthesis processes that can become a problem for the environment, even when originally not toxic. Future compound production systems, such as heterologous production in microorganisms, are in progress, but the number of suitable microorganisms and their culture optimization for biosynthesis is still limited. These methods have, however, the potential to allow for short transportation routes as production sites that are independent from special climate and growth conditions. Genome-engineered bacteria, as well as wild-type fungi [56], are expected to replace or supplement

plant culture systems for natural product production in the future. Genome-engineered microorganisms are, therefore, a key concept for further progress in the field. As stated by Wawrosch and Zotchev [57], the progress of methods for higher productivity of in vitro systems can presently not be found. For production via plant culture systems, a better understanding of genetic, biochemical, and physiological features leading to high metabolite accumulation is important and should inform future research. For instance, the roles of phytohormones other than jasmonate and methyljasmonate should be better evaluated. These investigations present another key concept for research in conventional and tissue culture. Conventional culturing and in situ cultivations must have continuance for the maintenance of plant endophytes and for plant fitness and evolution strengthened by environmental/microbial interactions. Table 1 summarizes the progress in production techniques to optimize the yields of sweet tasting molecules found in the presented plants.

**Table 1.** Progress in sweet tasting molecule production techniques.

| Plant Species Sweet Small Molecule | Culture and (Biotechnology-Based) Production Systems | | |
| :---: | :---: | :---: | :---: |
| | Conventional | In Vitro | via GMOs |
| *Siraitia grosvenorii* mogrol | - not established | - temporary immersion of single-node microcuttings [10] | - |
| *Hydrangea macrophylla* phyllodulcin | - to establish for Phyllodulcin elite cultivars [3] | - not established | - |
| *Glycyrrhiza glabra* glycyrrhizin | - not established for large scale production | - tissue culture using intermediate nodal explants [29,30] <br> - hairy root culture method [7] | - *S. cerevisiae* [32] |
| *Stevia rebaudiana* Steviol glycosides | - unsuited | - temporary immersion of nodal explants [33] | - *E. coli* [36] |

Beyond sustainable production, the contamination of the environment with sugar substitutes due to increasing use by humans is an increasing problem. In China, the artificial sweeteners acesulfame, sucralose, cyclamate, and saccharin were detected in ground-, surface, and drinking water, especially in winter time, which led to the search for methods to eliminate the contaminants [56,58]. Yang et al. [59] found sucralose as a persistent contaminant in the urban water cycle. The fact that artificial sweeteners occur as contaminates of water bodies has long been known [60,61].

Regarding the natural sweet tasting molecules, triterpenoid compounds are, at least to a considerable degree, resistant to natural degradation. Thus, there is a risk that these compounds could emerge as permanent contaminants in the water cycle as well. In contrast, phyllodulcin, which is obtained from a plant cultivable in temperate climates, seems to be degradable by long-known pathways that exist in many bacteria, although bibenzyl cleavage, for instance, of 2-[2-(3,4-dihydroxyphenyl)ethyl]-6-hydroxybenzoic acid as a starter reaction for further degradation is yet to be proved. Steviol and related compounds are degradable by microorganisms that exist in the natural habitats of *Stevia rebaudiana* in South America.

**Author Contributions:** M.P., A.U. and M.S. contributed equally to the writing, design, and implementation of the review. All authors have read and agreed to the published version of the manuscript.

**Funding:** Indirect financial support to A.U. and M.P. by the BMEL via Fachagentur für Nachwachsende Rohstoffe e.V. (FNR, grant number 22022617) is gratefully acknowledged.

**Institutional Review Board Statement:** Not relevant.

**Informed Consent Statement:** Not applicable.

**Data Availability Statement:** Not applicable.

**Acknowledgments:** We thank J. Ley (Symrise AG) for valuable comments and critical reading of the manuscript.

**Conflicts of Interest:** The authors declare no conflict of interest.

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
