# Peer review of "Culturing Important Plants for Sweet Secondary Products under Consideration of Environmentally Friendly Aspects"

_processes, doi:10.3390/pr10040703_

Round 1
Reviewer 1 Report
Dear authors, even though the argument is interesting and relevant, I cannot accept this review for publication because it does not include a methodology. Please consider adopting a systematic review methodology and try to submit again the manuscript.
Author Response
According to the instructions for authors also other forms of reviews can be submitted (Reviews: These provide concise and precise updates on the latest progress made in a given area of research. Systematic reviews should follow the PRISMA guidelines.) It was not our intention to submit a systematic review. Nevertheless, we changed the structure of the minireview and hope it will meet your understanding. Thank you for the notice.
The old structure of the minireview considered the different plants as units.
Reviewer 2 Report
Authors give an actual report on plants for sweet secondary products, including some information of their own research. Information is given on cultivation conditions, biosynthesis and degradation conditions. Furthermore economically and environmentally aspects of compatible usage are discussed.
The paper is written in good scientific and readable style. A few hints for slight improvements are indicated directly in the attached reviewed text.
The paper documents well the actual state of research in this field. It could be of importance for furhter research as well as for food producers. It should be published after minor revision.

Author Response
The authors thank the reviewer for the nice comment.
Reviewer 3 Report
The article presents a mini-review of cultivation regimes of mogrosides (Siraitia grosvenorii), phyllodulcin (Hydrangea macrophylla), glycyrrhizin (Glycyrrhiza glabra), steviol glycosides (Stevia grosvenorii) and rubusoside (Rubus suavissimus) with potentially economically significant sweet-tasting. Although, the article has an unstandardized structure. I have the following specific comments for the authors:
Thoroughly check the English in the article; there are numerous mistakes.
Page 1, In abstract, please avoid using a personal noun, instead of: we focus here on, the use-the focus was…We shortly expand, use -it was shorty expand…
Page 1, line 35, please, specify the natural sweet-tasting molecules;
Page 1, line 40, please avoid personal noun;
Page 3, Figure1, please leave just the caption of the figure, place the explanation of the biosynthesis in the text below and remove it from the caption;
Page 6, line 221, add a reference for precursor feeding studies;
Page 8, line 284, please specify in vitro propagation methods;
Page 8, line 315, Figure 6, please leave just the caption of the figure, place the explanation of the biosynthesis in the text below and remove it from the caption;
Page 10, line 389, Figure 7, please leave just the caption of the figure, place the explanation of the biosynthesis and degradation in the text below and remove it from the caption;
Page 11, please avoid using references in conclusion;
Author Response
The authors thank the reviewer for the helpful comments.
The structure of the article was changed.
Thoroughly check the English in the article; there are numerous mistakes. Done, but if not sufficient we will take advantage of the MDPI language service.
Page 1, In abstract, please avoid using a personal noun, instead of: we focus here on, the use-the focus was…We shortly expand, use -it was shorty expand… Done
Page 1, line 35, please, specify the natural sweet-tasting molecules; done
Page 1, line 40, please avoid personal noun; done
Page 3, Figure1, please leave just the caption of the figure, place the explanation of the biosynthesis in the text below and remove it from the caption; done
Page 6, line 221, add a reference for precursor feeding studies; After the next sentence, belonging to the part phyllodulcin biosynthesis, reference [46] is given, which presents the precursor studies (46. Eckermann, E.; Schröder, G.; Eckermann, S.; Strack, D.; Schmidt, J.; Schneider, B.; Schröder, J. Stilbenecarboxylate biosynthesis: a new function in the family of chalcone synthase-related proteins. Phytochemistry, 2003, 62, 271-286.).
Page 8, line 284, please specify in vitro propagation methods; We give more information to most recent methods.
Page 8, line 315, Figure 6, please leave just the caption of the figure, place the explanation of the biosynthesis in the text below and remove it from the caption; Done
Page 10, line 389, Figure 7, please leave just the caption of the figure, place the explanation of the biosynthesis and degradation in the text below and remove it from the caption; Done
Page 11, please avoid using references in conclusion; Changed in Discussion and Conclusion as did other authors in Processes.
Round 2
Reviewer 1 Report
Dear Authors,
the authors' instruction does not give indications on minireviews. So I leave it to the editors to decide on this issue.
From a technical point of view, a minireview should include:
1. A brief perspective and this is your introduction;
2. A summary of the established principles, and current state of the art, and sections 2, 3, 4, fulfil this requirement;
3. Highlight of future directions
4. Key concepts
The last two points are missing. So I kindly ask you, in the introduction, to state, given the importance of sweeteners for future markets, the importance of this research in identifying gaps and addressing future research directions. In the discussion, you have to add research gaps and suggest future research directions.
Please use the track change function while reviewing your paper.
Author Response
We were asked to include “Highlight of future directions and key concepts, to identify research gaps and further research directions”. We added such points to the text (introduction, discussion and conclusion) and hope we fulfilled the reviewer´s expectations. If we get a positive response, we will contact the MPDI language service for corrections.
Reviewer 3 Report
The author's responses are fair. The article can be accepted in its present form.
Author Response
The authors thank reviewer 3 again for the valuable comments and for the suggestion to accept the manuscript in the present form.
Round 3
Reviewer 1 Report
thank you, it is better now!